# Women’s Intentions to Engage in Risk-Reducing Behaviours after Receiving Personal Ovarian Cancer Risk Information: An Experimental Survey Study

**DOI:** 10.3390/cancers12123543

**Published:** 2020-11-27

**Authors:** Ailish Gallagher, Jo Waller, Ranjit Manchanda, Ian Jacobs, Saskia Sanderson

**Affiliations:** 1Research Department of Behavioural Science and Health, University College London, Gower Street, London WC1E 6BT, UK; Ailish.gallagher@gstt.nhs.uk; 2Cancer Prevention Group, School of Cancer & Pharmaceutical Sciences, King’s College London, Guy’s Hospital, Great Maze Pond, London SE1 9RT, UK; jo.waller@kcl.ac.uk; 3Wolfson Institute of Preventive Medicine, Barts Cancer Institute, Queen Mary University of London, Charterhouse Square, London EC1M 6BQ, UK; r.manchanda@qmul.ac.uk; 4Department of Gynaecological Oncology, Barts Health NHS Trust, London EC1A 7BE, UK; 5Department of Women’s Health, University of New South Wales, Australia, Level 1, Chancellery Building, Sydney 2052, Australia; i.jacobs@unsw.edu.au; 6Early Disease Detection Research Project UK (EDDRP UK), 2 Redman Place, London E20 1JQ, UK

**Keywords:** risk stratification, genomics, questionnaires, attitudes

## Abstract

**Simple Summary:**

Risk stratification using genetic testing to identify women at increased risk of ovarian cancer may increase the number of patients to whom risk-reducing surgery (e.g., salpingo-oophorectomy) may be offered. However, little is known about public acceptability of such approaches. Our online experimental survey aimed to explore whether women aged 45–75 in the general population are willing to undergo ovarian cancer risk assessment, including genetic testing, and whether women’s potential acceptance of risk-reducing surgery differs depending on their estimated risk. We looked at whether psychological and cognitive factors mediated women’s decision-making. The majority of participants would be interested in having genetic testing. In response to our hypothetical scenarios, a substantial proportion of participants were open to the idea of surgery to reduce risk of ovarian cancer, even if their absolute lifetime risk is only increased from 2% to 5 or 10%.

**Abstract:**

Risk stratification using genetic and/or other types of information could identify women at increased ovarian cancer risk. The aim of this study was to examine women’s potential reactions to ovarian cancer risk stratification. A total of 1017 women aged 45–75 years took part in an online experimental survey. Women were randomly assigned to one of three experimental conditions describing hypothetical personal results from ovarian cancer risk stratification, and asked to imagine they had received one of three results: (a) 5% lifetime risk due to single nucleotide polymorphisms (SNPs) and lifestyle factors; (b) 10% lifetime risk due to SNPs and lifestyle factors; (c) 10% lifetime risk due to a single rare mutation in a gene. Results: 83% of women indicated interest in having ovarian cancer risk assessment. After receiving their hypothetical risk estimates, 29% of women stated they would have risk-reducing surgery. Choosing risk-reducing surgery over other behavioural responses was associated with having higher surgery self-efficacy and perceived response-efficacy, but not with perceptions of disease threat, i.e., perceived risk or severity, or with experimental condition. A substantial proportion of women age 45–75 years may be open to the idea of surgery to reduce risk of ovarian cancer, even if their absolute lifetime risk is only increased to as little as 5 or 10%.

## 1. Introduction

Ovarian cancer is the sixth most common cancer among women in the UK. The general population lifetime risk of developing ovarian cancer is approximately 2%, and incidence is predicted to rise by 26% in the UK, 14% in Europe, and by 55% worldwide over the next two decades [1]. The risk of ovarian cancer rises with age, increasing significantly in women over 45 years [2]. DNA variants in a number of cancer susceptibility genes are known to be associated with ovarian cancer: women with a high penetrance genetic variant, such as a *BRCA1* or *BRCA2* mutation, are considered to be at high risk for developing breast and ovarian cancer [3,4,5]. Historically, genetic testing for ovarian cancer risk has been clinically indicated only for women with a strong family history of breast and/or ovarian cancer. However, using a family history based approach misses over half the cancer susceptibility gene (CSG) carriers at risk [6,7], and is associated with restricted access and limited utilization of genetic testing [8]. Additionally, the majority of cases of ovarian cancer do not occur in affected families [9]. There is increasing interest in the idea of adopting a risk-stratified approach to ovarian cancer prevention by offering genetic testing to all women regardless of family history [10,11,12].

In addition to rare variants of high penetrance, genome-wide association studies have to date identified a number of common single nucleotide polymorphisms (SNPs) associated with slightly increased risk of ovarian cancer [13]. SNP-based information and certain lifestyle factors each increase ovarian cancer risk by a small amount individually, but this becomes clinically significant when the information is combined, e.g., from 2% to between 5% and 10% lifetime risk [9,14,15]. Surgical prevention has been shown to be cost-effective at the 4–5% ovarian cancer risk threshold [16,17]. Newer risk models and recently validated intermediate risk genes can identify individuals at these risk thresholds. Risk stratification using multigene testing to identify women at increased risk of ovarian cancer is potentially more cost- and time-effective than single gene testing and increases the number of patients to whom risk-reducing surgery (e.g., salpingo-oophorectomy) may be offered [18]. While clinical practice has gradually begun to change [19], data on public acceptability of such approaches are limited.

An initial quantitative study assessing attitudes towards population-based genetic testing for ovarian cancer risk in a general population sample found high levels of support for risk-stratified ovarian cancer screening based on prior genetic risk assessment [20]. There is good evidence to suggest that population-wide genetic testing for ovarian cancer is acceptable, feasible and cost-effective amongst Ashkenazi Jewish populations [6,7,21,22,23]. Preliminary data from the general population also indicate that population-based personalised ovarian cancer risk stratification is feasible, acceptable, has high satisfaction, reduces cancer worry/risk perception, and does not negatively impact psychological health or quality of life [12].

Bilateral risk-reducing salpingo-oophorectomy (surgical removal of the ovaries and fallopian tubes, hereafter referred to as “risk-reducing salpingo-oophorectomy” or “RRSO”) is currently recommended as the main and most effective preventative strategy for ovarian cancer in women at increased risk of ovarian cancer such as *BRCA* mutation carriers. RRSO can reduce ovarian cancer risk by 85–90% [24]. Traditionally the most common group of women undergoing surgical prevention have been *BRCA* carriers, who have a 17–44% lifetime risk of ovarian cancer [5,25]. In the UK, women with an estimated lifetime ovarian cancer risk of greater than 10%, who have completed their families, have traditionally been offered risk-reducing surgery [15]. Undertaking surgery on the basis of family history alone in the absence of a known mutation (at lower than *BRCA* levels of risk) has thus been clinical practice in the UK and other countries for many years [25,26]. Recently, the 10% threshold was relaxed to 4–5% [14,15]. A number of new ovarian cancer risk genes have been identified, such as *RAD51C* (lifetime risk 11%) [27], *RAD51D* (lifetime risk 13%), *PALB2* (lifetime risk 5%) [28], and *BRIP1* (lifetime risk 5.8%) [29], testing for which is part of routine clinical practice. RRSO is now offered and being undertaken for these CSGs too. Thus a number of clinical teams now offer RRSO to women in the “intermediate” risk category (5–10%) as well as those in the “high” risk category (over 10%) [15]. Additionally, more complex models using SNP profiles, in combination with other epidemiological and genetic risk factors, are being validated, which will provide absolute lifetime risk estimates in these ranges in the not too distant future [12].

National screening programmes for ovarian cancer are unavailable. In a large randomised control trial designed to establish the effect of early detection by ovarian screening in the general low-risk population, no conclusive significant impact on mortality from ovarian cancer was found [30], and definitive mortality data are awaited in 2021. Surveillance for those identified as high-risk (or in some cases moderate-to-high-risk) for ovarian cancer currently consists of serial 3–4 monthly serum CA125 (Cancer Antigen 125 protein; a tumour marker) measurement (and annual transvaginal ultrasound) aiming to detect pre-symptomatic cancer in the earlier stages and/or low volume disease where treatment is more effective [31]. This 4 monthly longitudinal CA125 biomarker driven surveillance strategy, using the risk of the ovarian cancer (ROCA) algorithm, may be beneficial in women at high risk of ovarian cancer [31]. We have shown that this is associated with a significant stage shift, which can be a surrogate for improved survival [31]. Identifying those at increased risk using a population wide risk-stratified approach may result in more timely risk reduction options and could have a significant impact on disease burden: modelling suggests that 13% of the female UK population have greater than 4% lifetime risk and 9% have greater than 5% lifetime risk [15]. Manchanda et al. (2018) suggest that, based on National Institute for Health and Care Excellence (NICE) cost-effectiveness guidelines, risk-reducing surgery may be cost effective for postmenopausal women over the age of 50, with a lifetime ovarian cancer risk of ≥5%. Wider implementation of targeted surgical prevention for women at greater than 4–5% lifetime risk threshold provides a huge opportunity for cost-effective targeted primary prevention.

Offering risk stratification to women in the general population, including communicating personal ovarian cancer risk information and offering risk-reducing surgery, has the potential to be a feasible way to reduce ovarian cancer mortality and reduce the population burden of the disease. However, risk stratification will only lead to improved ovarian cancer prevention and early diagnosis if women whose results indicate increased risk take action to reduce their risk. Women with a family history of breast and ovarian cancer have been found to opt for risk reduction surgery, e.g., among *BRCA1* and *BRCA2* mutation carriers, the majority underwent risk-reducing surgery (salpingo-oophorectomy) after their risk was communicated to them [25,32]. However, although quite a lot is known about how genetic risk information impacts psychological wellbeing and behaviours among women from families affected with ovarian (and/or breast) cancer, less is known about how women in the wider non-Jewish population might react to being informed they have an increased genetic risk of ovarian cancer [33,34,35,36,37]. Further research is needed to determine how women in the general population might respond if presented with ovarian cancer risk information indicating they are at high risk based on genetic as well as other risk factors.

Based on research prior to 2016, the evidence does not support the hypothesis that communicating CSG-based risk estimates motivates lifestyle behaviour changes [33,38]. CSG-based risk information also has not been associated with negative psychological outcomes [7,33,38,39]. More recently, a nested study within the Predicting Risk of Cancer at Screening (PROCAS) study was conducted comparing the psychological impact of providing women with personalised breast cancer risk estimates based on: (a) the Tyrer–Cuzick (T–C) risk algorithm including breast density, or (b) T–C including breast density plus SNPs, versus (c) comparison women awaiting results. This study found little evidence of either psychological harm or of differences between women provided with risk estimates based on SNPs versus others. However, women categorised as high-risk were excluded from the study, so no conclusions could be drawn regarding high-risk results specifically. It remains to be seen whether the source of the risk may have impacted psychological factors or if it had an effect on acceptance of the risk information in this study [40]. In another recent study that examined the impact of returning secondary findings (including *BRCA1/2*) from genomic sequencing to unselected populations, few adverse psychological effects were found [41].

As an initial step to providing some empirical data on the question of how women in the general population might respond to personal ovarian cancer risk information indicating increased risk (as against moderate risk [40]), we conducted an experimental survey study with women in the general population, using the Extended Parallel Process Model (EPPM) [42] as our theoretical framework, and to inform our selection of variables and measures. The EPPM is a social cognition model of information processing and behaviour: it posits that how individuals react to threatening information is informed by (a) their perceptions of the threat (perceived risk or susceptibility, and perceived severity), and (b) their perceptions of the recommended action to reduce the threat (self-efficacy, i.e., their confidence in their ability to carry out the recommended behaviour, and perceived response efficacy, i.e., their confidence that the recommended behaviour will effectively reduce the threat to their health).

Our specific aims were to: (1) explore whether women in the general population are willing to undergo ovarian cancer risk assessment which includes genetic testing; (2) examine whether women’s potential acceptance of risk-reducing surgery differs depending on whether their estimated risk is 5% or 10%; (3) examine whether women’s potential acceptance of risk-reducing surgery differs depending on whether their estimated risk is based on a single rare genetic variant of high penetrance or a more complex combination of genetic and non-genetic factors. We also explored whether threat and efficacy cognitions mediated any observed between-group differences, and examined the associations between these cognitions (threat, efficacy) and acceptance of risk-reducing surgery in the sample overall.

## 2. Results

### 2.1. Sample Characteristics

Table 1 provides an overview of the participant characteristics. Age ranged from 45 to 75 years with a mean of 57.50 (SD = 8.13). The majority were White (95.6%) with 3.8% from other ethnic backgrounds. Educational attainment was fairly evenly split between General Certificate of Secondary Education (GCSE) or equivalent (34.6%), A levels or equivalent (23.8%), and undergraduate degree or equivalent (24.1%). The majority (85.2%) of women were either perimenopausal (beginning menopause) or post-menopausal. See Figure 1 for the Consolidated Standards of Reporting Trials (CONSORT) flow diagram of participants throughout the study.

### 2.2. Interest in Ovarian Cancer Risk Assessment

Overall, 83.2% of women indicated they would “yes definitely” (38.0%) or “yes probably” (45.2%) have an ovarian cancer risk assessment if it was offered to them by their general practitioner (GP) on the National Health Service (NHS) (see Table 1 and Figure 2).

### 2.3. Behavioural Response to Personalised Ovarian Cancer Risk Information

After receiving their hypothetical risk result, 28.5% of women said they would opt for risk-reducing surgery, 33.9% for increased surveillance (transvaginal ultrasound), and 20.9% would make lifestyle changes (e.g., quitting smoking, maintaining a healthy weight; see Figure 3 and Appendix A).

### 2.4. Differences by Experimental Condition

Women’s intentions to have risk-reducing surgery did not differ significantly between the 5% and 10% multifactorial SNPs + lifestyle groups (27.9% vs. 26.2%, respectively) (χ^2^(1) = 0.314, *p* = 0.61). Women who received a 10% risk result based on a rare genetic variant were no more likely to opt for RRSO over other risk-reducing options than women who received a 10% risk result based on multifactorial SNPs + lifestyle factors (31.4% vs. 26.2%, respectively) (χ^2^(1) = 2.512, *p* = 0.13).

### 2.5. EPPM Variables

The mean (M) and standard deviation (SD) of the EPPM variables were: perceived risk (M = 3.51, SD = 0.82), perceived severity (M = 4.52, SD = 0.58), perceived response-efficacy (M = 4.03, SD = 0.78), perceived self-efficacy (M = 2.98, SD = 1.34). Means by exposure group are shown in the Appendix A).

### 2.6. Intention to Have Risk-Reducing Surgery (RRSO) versus Other Risk-Management Options

A binary logistic regression was conducted to investigate what factors were associated with hypothetical intention to have risk-reducing surgery vs. other behavioural options. Independent variables included in the model were age, ethnicity, educational attainment, previous breast and cervical screening participation, experimental group and EPPM variables (perceived risk, perceived severity, self-efficacy, perceived response-efficacy). In unadjusted analyses, women reporting higher perceived risk of ovarian cancer and higher perceived severity of ovarian cancer (i.e., the perceived threat variables), and higher surgery self-efficacy and perceived response-efficacy (i.e., variables relating to perceptions of the risk-reducing behaviour) were more likely than other women to opt for risk-reducing surgery. In the multivariable model, perceived response-efficacy (odds ratio (OR) = 2.22; 95% confidence interval (CI): 1.64–3.00) and self-efficacy (OR = 1.90; 95% CI: 1.63–2.22) remained significantly associated, whereas the perceived threat variables were no longer significantly associated, with choosing risk-reducing surgery over other behavioural options. None of the measured socio-demographic or health-related factors were significantly associated with intention to have surgery (see Table 2).

## 3. Discussion

A high proportion (83%) of women in this sample indicated they would be interested in having an ovarian cancer risk assessment if offered by their GP on the NHS. This is consistent with previous research by Meisel et al. (2016), who found that 88% of a general population sample of women in the UK would be interested in genetic testing for ovarian cancer risk if it were offered by the NHS, and included information about breast cancer risk, echoing previous support from qualitative research for the availability of genetic testing and risk-stratified screening [43]. It is also consistent with uptake of genetic testing in our population-based studies [12,21].

We also found that a substantial proportion of British women over the age of 45 years might be open to the idea of having RRSO, even if their absolute lifetime risk were increased from a general population risk of 2% to as little as 5% or 10%. In addition, we also demonstrated in multivariable analyses that perceptions of risk-reducing surgery (self-efficacy and perceived response-efficacy) were independently associated with choosing surgery over other options, whereas the perceived threat of ovarian cancer (perceived risk and perceived severity) was not.

Although over a quarter (29%) of women opted for RRSO, slightly more women opted for surveillance (34%). The observed preference for surveillance may be due to the invasiveness of surgery, and could also potentially be due to the generally positive attitude towards cancer screening in the UK [44]. Lack of detailed information on the efficacy of each risk management option due to the hypothetical nature of this study may have resulted in participants deciding on the less invasive option, i.e., surveillance. Research suggests perceptions about risk-reducing surgery and surveillance are potentially modifiable: Mai et al. (2017) identified misperceptions about ovarian cancer risk and benefits of screening as important factors influencing decisions about risk-reducing surgery versus surveillance [45]. The concept of common genetic variants of low penetrance single nucleotide polymorphisms (SNPs) may be unfamiliar to the majority of individuals in the general public. For example, in a study by French et al. (2018), there was considerable variation in understanding of test results. The role of SNPs in cancer risk may be less familiar to individuals than more widely publicised rare genetic variants such as those in the *BRCA* genes [40]. Additionally, lifestyle factors may be perceived as being under greater personal control and, therefore, less serious than rare genetic variants.

Our study findings suggest that individuals interpreted the two levels of risk (5% vs. 10%) similarly, with the difference in communicated risk having a non-significant impact on participants’ intentions to have RRSO. This supports previous research exploring the effect of risk information on behaviour, suggesting there is not a simple linear relationship between increments in risk and risk perception [39,40].

In addition to the lack of impact on perceived risk of ovarian cancer, we similarly found that different presentations of risk in the hypothetical scenarios (5% SNPs + lifestyle vs. 10% SNPs + lifestyle risk; 10% rare genetic variant vs. 10% SNPs + lifestyle) did not lead to differences in any other cognitive factors considered in the EPPM framework (i.e., perceived severity of ovarian cancer, perceived response-efficacy of risk-reducing surgery, self-efficacy to undertake risk-reducing surgery).

In contrast, we found that, in the sample overall, opting for risk-reducing surgery was associated with higher self-efficacy and higher perceived response-efficacy of risk-reducing surgery. According to the Extended Parallel Processing Model [46], higher perceptions of self-efficacy and/or response-efficacy relating to the recommended behaviour are associated with greater likelihood that systematic processing of threatening (risk) information will occur. Conversely, when perceptions of efficacy are low, people are more likely to avoid threatening risk information. Together, our findings suggest that if women perceive or believe that RRSO is being recommended to them clinically, this may have a greater impact on their decision-making than the details of their risk result (i.e., whether their risk is 5% or 10%, and whether that risk is based on a single rare genetic variant or a more complex combination of SNPs and lifestyle factors). The observation that psychological variables had a greater impact on intentions than the absolute risk numbers suggest that this might be important to consider in any potential future national rollouts. Offering psychological support for those who need it as part of the RRSO discussion and decision-making process is part of routine clinical practice in many centres today. Our study highlights the importance of incorporating this into future national guidelines.

In previous research using hypothetical scenarios, there has been some evidence to suggest that genetic information leads to more deterministic responses than non-genetic information [33,38,47,48,49]. Our study did not include a non-genetic condition and so does not speak to this aspect of how people respond to personal genetic versus non-genetic information.

The present study had several limitations. The cross-sectional design of the study did not allow for causation to be inferred. However, exploratory experimental studies such as this one can be valuable in informing hypotheses before moving on to study designs designed to trial real risk assessments. The use of hypothetical scenarios was both a strength and a limitation. This study attempted to model a “real life” scenario in which genetic risk information was provided to the general population. However, many of the contextual details and additional resources that accompany risk information are not available in hypothetical scenarios, which may limit the ecological validity of the study. This study was concerned with behavioural intention, as ovarian cancer population surveillance or population-wide genetic testing is not currently clinically available, so actual behaviour could not be measured. The presence of a potential intention behaviour gap is well established for other clinical interventions and cannot be excluded here.

The measures used in this study are adapted from previous research; however, they were almost all single-item measures, which may not be sensitive enough to adequately represent the underlying construct being measured due to the lack of psychometric information (e.g., test-retest reliability, discriminant or convergent validity). Prior knowledge may have an influence on how individuals appraise threatening health information [50]: this study did not measure previous ovarian cancer and genetic risk knowledge or previous genetic testing, which may have had an impact on behavioural intention (however this may be unlikely given this type of genetic testing is not widely available in the UK). In addition, we did not assess understanding of the information provided, and it is possible that some concepts (e.g., SNPs) may not have been well understood.

The sample was predominately White British and, therefore, may not be generalisable to other ethnic groups, given decisions about risk-reducing surgery and psychological effects may differ cross-culturally. In addition, the restricted age-range of the sample limits the generalisability of the findings (e.g., to younger women age 35 years and over who may also be offered risk-reducing surgery if they are at high risk). However, most women from the general population who are at increased risk of ovarian cancer will fall in the intermediate risk (5–10% lifetime risk) category [9]. RRSO at intermediate ovarian cancer risk levels (including for moderate penetrance CSGs) is recommended to be undertaken over the age of 45–50 years [15]. The sample was self-selected and may have had greater interest in the topic than the wider general population; the generalizability of the findings is therefore uncertain. Finally, as with any experimental study, we are unable to rule out the possibility of demand characteristics, i.e., participants responding in a way they think is expected according to their perceptions of the aim of the study. Despite the limitations, this study provides insights on the effects of experimentally manipulating genetic risk information for ovarian cancer on outcomes, comparing different sources and levels of risk on risk management behavioural intentions and psychological variables in the general population.

The information provided before being exposed to the hypothetical risk scenario on ovarian cancer, risk factors, and the efficacy of risk management was necessarily basic and brief, which may be an additional limitation. However, previous research did not find any difference in behavioural outcomes between use of gist and extended versions of decision aids in relation to ovarian cancer risk management [51]. Future research might usefully provide more detailed information containing details about the efficacy of a particular risk management behaviour to encourage “danger control” cognitive processing. This may aid in changing risk management preferences. 

Future research might also benefit from including measures of other psychological and cognitive factors as potential predictors of risk management, e.g., causal beliefs. In addition, further research is needed on communicating risk information incorporating genetics to people in the general population outside of traditional clinical genetics department settings. Furthermore, a control group where participants are given a general population-based risk estimate would be useful for future research, as we were unable to compare between the general population risk and increased risk in this study.

The mean age of participants in this study was 57 years, with the majority of participants having completed having children and/or being past childbearing age, with most participants reporting they had begun menopause or were post-menopausal. Future research should explore the psychological and cognitive effects of ovarian cancer risk information being offered to younger women. In addition, there were relatively few women in the oldest age group (71–75 years) in this study, and so it is possible the apparent trend of increasing age being less associated with interest in surgery was due to the study being underpowered. This could also be a topic of investigation in future research.

Risk-reducing surgery, specifically RRSO, is at present the most effective risk management option available to women at increased risk of ovarian cancer. Our findings suggest there are a number of cognitive factors that influence intention to have ovarian cancer risk-reducing clinical interventions, beyond perceptions of risk. Future research should explore other possible factors that may have an impact on decision-making about risk management strategies. It is imperative to identify whether and/or how genetic risk information about common complex diseases will be translated into public health benefit: this is arguably especially urgent for diseases, such as ovarian cancer, which are characterised by being notoriously difficult to detect early and by having a high prevalence of late-stage diagnosis. Combined testing for multiple genetic factors together with lifestyle and other risk factors may lead to the ability to stratify the population for ovarian cancer risk for targeted prevention thus potentially saving lives.

Population testing provides a new paradigm for ovarian cancer prevention and can prevent thousands more cancers than the current clinical approach [52]. Jewish population studies support population testing for CSGs [53]. Our pilot study shows that population testing for lifetime risk of ovarian cancer is feasible, acceptable and has high satisfaction in general population women [12]. However, there is now need for large implementation studies, with long term outcomes, to provide real world evidence and develop context-specific models for implementing this approach for women in the general population. This will valuably inform future policy decisions regarding population-wide risk stratified approaches for risk-adapted ovarian cancer screening and prevention.

## 4. Materials and Methods

### 4.1. Overview

Participants (*n* = 1017) were women aged 45–75 years recruited via online survey company Survey Sampling International (SSI) in July 2017. An email containing a web link was sent to SSI panellists who fit the study criteria with respect to gender and age, inviting them to take part. The email did not contain information about the topic of the study. Those responding were directed to a short screening questionnaire. Eligible participants were then presented with a consent form. Incentive points, which can be exchanged for shopping vouchers, were awarded to SSI panellists for their time (equivalent to ~£0.50 for this 10 min study).

All participants were given information about ovarian cancer, including that on average around 2% of women will develop ovarian cancer in their lifetime; asked to imagine that they had had an ovarian cancer risk assessment via the NHS; and asked to imagine they had received a result indicating they were at increased risk of ovarian cancer (see Appendix B).

Women were randomly assigned to one of three experimental conditions using a software algorithm (See Figure 1). They were asked to imagine they had undergone an ovarian cancer risk assessment and had received this personalised risk estimate from their GP: (a) 5% ovarian cancer risk due to common genetic variants and lifestyle factors; (b) 10% ovarian cancer risk due to common genetic variants and lifestyle factors; or (c) 10% ovarian cancer risk due to a single rare variant in a cancer susceptibility gene such as *BRCA2* (see Appendix C).

The study was approved by the University College London ethics committee (Project ID Number: 10251/001).

### 4.2. Inclusion Criteria

Eligible participants were women aged 45–75 years, with no previous history of breast or ovarian cancer diagnosis. Women who indicated they were unsure of, or had not completed childbearing, were excluded from analyses (*n* = 13).

### 4.3. Measures

All measures are shown in Appendix D.

#### 4.3.1. Interest in Ovarian Cancer Risk Assessment

Interest was assessed before exposure to the hypothetical test results, with the item, “If your GP offered you this ovarian cancer risk assessment on the NHS, would you take up the offer?” (adapted from [20]. Response options were “no, definitely not”; “no probably not”; “yes, probably” and “yes, definitely”. The information women read before answering the question explained that the risk assessment would involve providing lifestyle information as well as a blood sample for genetic testing (see Appendix B).

#### 4.3.2. Perceived Risk of Ovarian Cancer

Perceived risk was measured using a single item, “If I had just received this personal ovarian cancer risk result, I would feel that my risk of developing ovarian cancer was” adapted from [54]. Responses for these questions were recorded on a 5-point Likert scale with response options ranging from “much lower than other women of my age” to “much higher than women of my age”. A higher score on the 5-point scale indicated greater perceived risk.

#### 4.3.3. Perceived Severity of Ovarian Cancer

Perceived severity was measured using two questions adapted from [55]: “Developing ovarian cancer would have major consequences on my life” and “ovarian cancer is a serious condition” with five response options ranging from “strongly agree” to “strongly disagree”. A higher score on the (possible scores 1–5) scale indicated greater perceived severity.

#### 4.3.4. Self-Efficacy for Risk-Reducing Surgery

One item, adapted from [56], assessed participants’ confidence in their ability to have risk-reducing surgery. Individuals were asked “How confident are you that you would go through with risk-reducing surgery if you were motivated to do so”. The response options ranged from “not at all confident” to “extremely confident”. A higher score on the scale indicated greater perceived self-efficacy.

#### 4.3.5. Perceived Response-Efficacy of Risk-Reducing Surgery

For perceived response-efficacy of risk-reducing surgery, participants were asked to indicate how effective they felt risk-reducing surgery would be in lowering their ovarian cancer risk using a single item adapted from [56]: “Having surgery to remove your ovaries and fallopian tubes is an effective way to lower your risk of ovarian cancer”. The response options were “strongly agree” to “strongly disagree”. Items were reverse coded: a higher score on the (possible scores 1–5) scale indicated greater perceived response-efficacy.

#### 4.3.6. Behavioural Intention

To assess women’s potential behavioural reactions to their risk results, they were asked: “If I had just received this personal ovarian cancer risk result, I would choose to...”. The response options were: “have risk-reducing surgery to remove my ovaries”; “have surveillance such as regular ultrasound scans”; “make lifestyle changes”; “do nothing”; and “I am not sure what I would do”.

#### 4.3.7. Demographic and Health Characteristic Measures

Information on demographics was collected from all participants including: age, ethnicity, educational attainment, relationship status, health characteristics, family history of cancer, personal history of cancer, menopause status, and breast and cervical screening attendance. Ethnicity (White vs. other ethnic group), menopause status (pre-menopausal vs. peri/post-menopause) and breast and cervical screening attendance (regular vs. irregular or not yet eligible) were dichotomised.

### 4.4. Data Analyses

A power calculation based on the primary binary outcome, intention to have risk-reducing surgery, taking into account group comparisons of three groups, suggested a sample size of 782 was required (medium effect size, power of 90%, alpha of 0.05). All statistical analyses of the data were carried out using SPSS 24. Analyses of variance (ANOVAs) and chi-square tests were conducted to explore between-group differences. Logistic regression was used to explore predictors of willingness to have risk-reducing surgery (vs. other behavioural responses to the risk information). Unadjusted and adjusted models were examined to explore the predictive effect of the experimental group and psychological variables on intention to have surgery and address possible demographic and health-related covariates.

## 5. Conclusions

The findings of this study contribute to a growing body of risk stratification research exploring the potential usefulness and clinical utility of population-wide risk assessment incorporating genetic testing alongside other risk factors. The need for risk stratification is perhaps particularly urgent for diseases, such as ovarian cancer, where survival outcomes are poor and population-wide screening for the disease is not currently recommended. We provide initial evidence, suggesting that a substantial proportion of women aged 45 years and over are open to the idea of risk stratification and having surgery to reduce their risk of ovarian cancer in response to increased risk results, even if their absolute lifetime risk is only increased by a few percentage points in absolute terms. Our findings do not speak to other barriers that might prevent women’s behavioural intentions or preferences being translated into actions—barriers such as lack of timely access to healthcare services.

## Figures and Tables

**Figure 1 cancers-12-03543-f001:**
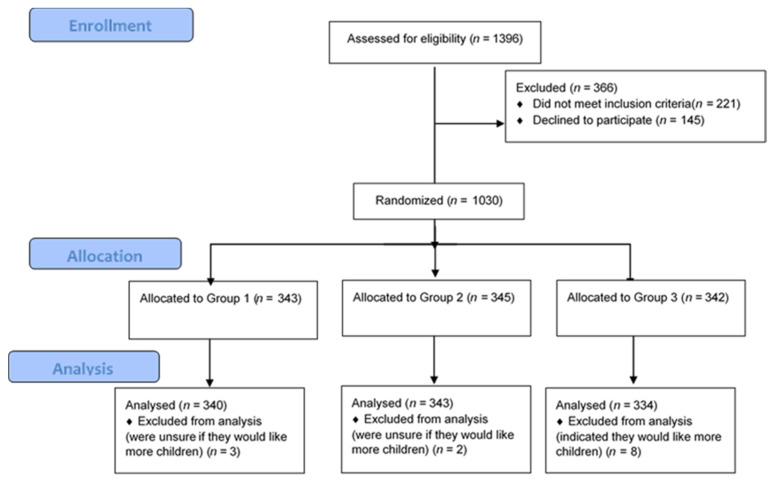
CONSORT 2010 Flow Diagram.

**Figure 2 cancers-12-03543-f002:**
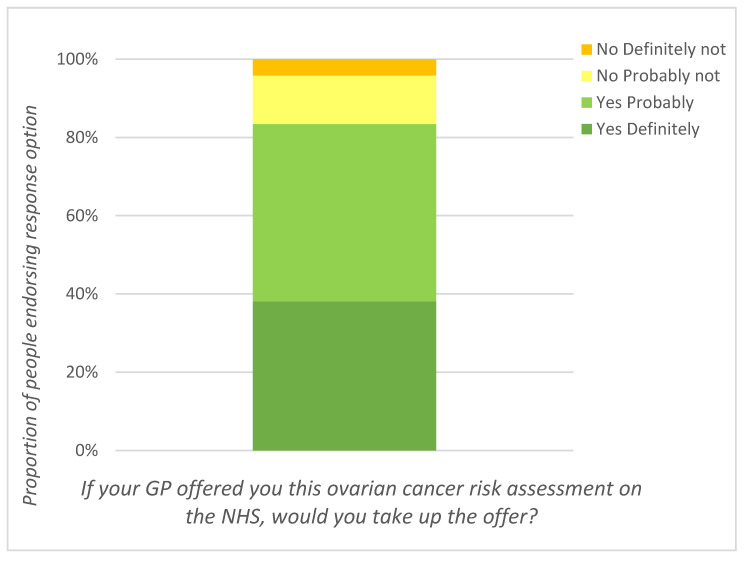
Interest in ovarian cancer risk assessment.

**Figure 3 cancers-12-03543-f003:**
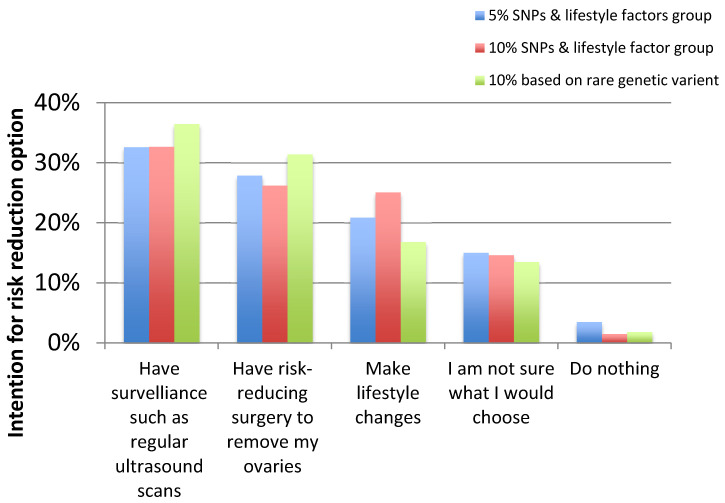
Behavioural intentions after exposure to hypothetical risk scenario compared between groups.

**Table 1 cancers-12-03543-t001:** Sample characteristics and interest in genetic testing overall and in each randomised experimental group (total *n* = 1017).

Variables	Group 1: 5% SNPs and Lifestyle (*n* = 340)	Group 2: 10% SNPs and Lifestyle (*n* = 343)	Group 3: 10% Rare Genetic Variant (*n* = 334)
Demographics	***n* (%)**		
Age Mean (SD)	57.43 (8.32)	57.37 (7.78)	58.08 (8.19)
Age group			
45–50	89 (26.2)	82 (23.9)	73 (21.9)
51–55	71 (20.9)	66 (19.2)	67 (20.1)
56–60	59 (17.4)	70 (20.4)	65 (19.5)
61–65	53 (15.6)	70 (20.4)	61 (18.3)
66–70	42 (12.4)	34 (9.9)	38 (11.4)
71–75	26 (7.6)	21 (6.1)	30 (9.0)
Ethnicity			
White (Any background)	327 (96.2)	324 (94.5)	321 (96.1)
Other ethnic group	11 (3.2)	17 (5.0)	11 (3.3)
Educational Attainment			
No Formal Qualification	26 (7.6)	23 (6.7)	17 (5.1)
GCSE or equivalent	115 (33.8)	126 (36.7)	111 (33.2)
A-Levels or equivalent	75 (22.1)	80 (23.3)	87 (26.0)
Undergraduate degree/equivalent	89 (26.2)	77 (22.4)	79 (23.7)
Postgraduate degree/equivalent	31 (9.1)	30 (8.7)	31 (9.3)
Other	4 (1.2)	7 (2.0)	9 (2.7)
Relationship Status			
Married/Cohabiting/In a relationship	245 (72.1)	232 (67.6)	233 (69.8)
Single/Separated/divorced/widowed	93 (27.4)	111 (32.4)	99 (29.6)
Health Characteristics			
Menopause status			
Premenopausal	41 (12.1)	37 (10.8)	43 (12.9)
During/post menopause	283 (83.2)	298 (86.9)	285 (85.3)
Personal History of Cancer			
Yes	17 (5.0)	16 (4.7)	17 (5.1)
No/Not sure	323 (95.0)	327 (95.3)	317 (94.9)
Family History of Cancer			
Yes	214 (62.4%)	206 (59.7%)	201 (58.8%)
No/Not sure	129 (37.6%)	139 (40.3%)	141 (41.2%)
Cervical Screening			
Regular	192 (72.7)	214 (76.7)	179 (67.8)
Irregular	72 (27.3)	65 (23.3)	85 (32.2)
Not eligible	58 (17.1)	44 (12.8)	62 (18.6)
Breast Screening			
Regular	196 (81.0)	201 (79.4)	196 (79.7)
Irregular	46 (19.0)	52 (20.6)	50 (20.3)
Not eligible	72 (21.2)	59 (17.2)	61 (18.3)
Interest in ovarian cancer risk assessment			
Yes Definitely	139 (40.9)	122 (35.6)	125 (37.4)
Yes Probably	151 (44.4)	156 (45.5)	153 (45.8)
No Probably not	36 (10.6)	46 (13.4)	46 (13.8)
No definitely not	14 (4.1)	19 (5.5)	10 (3.0)

SNPs: single nucleotide polymorphisms; SD: standard deviation; GCSE: General Certificate of Secondary Education.

**Table 2 cancers-12-03543-t002:** Logistic regression predicting likelihood of intending to have risk-reducing surgery vs. other behavioural response (*n* = 1017).

Variable	Intention to Have Risk-Reducing Surgery	Odds Ratios (95% CI)	
		Unadjusted	Adjusted
Demographic Factors	*n* (%)		
Age			
45–50	74 (30.3)	Ref	Ref
51–55	65 (31.9)	1.07 (0.72–1.61)	0.94 (0.51–1.74)
56–60	60 (30.9)	1.03 (0.68–1.55)	1.00 (0.53–1.87)
61–65	49 (26.6)	0.83 (0.55–1.28)	0.62 (0.32–1.19)
66–70	27 (23.7)	0.71 (0.43–1.19)	0.48 (0.14–1.65)
71–75	15 (19.5)	0.56 (0.30–1.04)	0.29 (0.06–1.50)
Ethnicity			
White (Any background)	278 (28.6)	Ref	Ref
Other ethnic group	11 (28.2)	0.98 (0.48–2.00)	1.16 (0.38–3.50)
Educational Attainment			
No formal qualifications	16 (24.2)	Ref	
GCSE/O Levels	111 (31.5)	1.44 (0.79–2.64)	
A-Levels or Equivalent	68 (28.1)	1.22 (0.68–2.29)	
Undergraduate degree	59 (24.1)	0.99 (0.53–1.87)	
Postgraduate degree	28 (30.4)	1.37 (0.67–2.80)	
Other	8 (40.0)	2.08 (0.72–6.00)	
Relationship Status			
Not married/in a relationship	82 (27.1)	Ref	
Married/in a relationship	207 (29.2)	1.11 (0.82–1.50)	
Health history			
Cervical Screening Attendance (*n* = 807)			
Regular	200 (34.2)	Ref	Ref
Irregular	49 (22.1)	0.59 (0.41–0.85) *	0.67 (0.41–1.11)
Breast Screening Attendance (*n* = 741)			
Regular	205 (34.6)	Ref	Ref
Irregular	35 (23.6)	0.53 (0.36–0.80) *	0.67 (0.38–1.17)
Menopause status			
Pre-menopause	35 (28.9)	Ref	Ref
Peri/Post-menopause	243 (28.1)	0.96 (0.63–1.46)	
Extended Parallel Processing Model Variables			
Perceived Risk		1.43 (1.19–1.71) **	1.14 (0.90–1.45)
Perceived Severity		1.42 (1.11–1.82) *	1.08 (0.74–1.57)
Self-Efficacy		2.19 (1.94–2.47) **	1.90 (1.63–2.22) **
Perceived Response Efficacy		3.15 (2.50–3.96) **	2.22 (1.64–3.00) **
Experimental condition			
5% SNPs & Lifestyle	95 (32.8)	Ref	Ref
10% SNPS & Lifestyle	90 (31.0)	0.92 (0.66–1.29)	1.08 (0.67–1.73)
10% rare genetic variant	105 (36.2)	1.18 (0.85–1.65)	1.87 (1.17–3.00) **

** Predictor significant at the 0.01 level (2-tailed), * Predictor significant at the 0.05 level (2-tailed), CI = Confidence Interval. Ref = reference category.

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
