# Peer review of "Women’s Intentions to Engage in Risk-Reducing Behaviours after Receiving Personal Ovarian Cancer Risk Information: An Experimental Survey Study"

_cancers, 2020, doi:10.3390/cancers12123543_

Round 1

Reviewer 1 Report

This hypothetical vignette study examined the reactions of women randomised to one of three hypothetical scenarios regarding communication of ovarian risk.  The study was generally clearly reported, and with appropriate conclusions drawn.

There are several issues that the authors might consider:

  1. Very strange having the Methods section after the Discussion – this should be moved.
  2. In the abstract, make clear the %ages are lifetime risk.
  3. In the background, it would be helpful to know if 5% to 10% were realistic clinical scenarios – these seem a bit low for e.g. BRCA2 mutations for a 45 year old woman.
  4. The sample is likely to be highly selective in terms of interest of this topic, so this caveat should be highlighted more in the discussion. More information on how the women were approached (e.g. where they on a bank of potential responders from the survey organisation, or where they recruited specifically for this study?  How much were they paid?)  On the positive side, there was a decent spread of educational achievement, which is often a weakness of studies using this method.
  5. There is also a related issue of demand characteristics of the study, which could be taken as implying that a response was appropriate.
  6. How strong was evidence that “SNPs” was understood?
  7. I would like to see the means for EPPM measures reported for each condition, and compared, e.g. was there a greater increase in response efficacy for the SNPs + lifestyle v rare mutation groups? That is, was attributing different cause related to how successful different treatment options were viewed?
  8. Were EPPM measures collected only for one group, i.e. are similar measures collected from other groups? And were these measures collected for surgery as an outcome only?  If data collected, it would be useful to see it presented.
  9. There appeared to be a trend of increasing age being less associated with surgery - this was apparently under-powered due to low numbers of older women. This could be further investigated or examined further.  Would 75 year old women really be considering surgery (or having it recommended?)
  10. Two implications that could be flagged up more. One, that %age risk was not important for surgery decisions, whereas EPPM variables were – this flags up the key importance of consideration of these psychological factors in any future rollout – as well as shows yet again that people struggle to make sense of absolute risks (is 10% good or bad?)
  11. Two, that the presentation of material as due to genetic factors v lifestyle appears important in affecting treatment decisions – which relates to the older Marteau vignette studies on this topic. It might be useful to relate the findings to this previous literature, which investigated attributions of cause and genetic determinism over many studies.

Author Response

Reviewer 1

This hypothetical vignette study examined the reactions of women randomised to one of three hypothetical scenarios regarding communication of ovarian risk.  The study was generally clearly reported, and with appropriate conclusions drawn.

There are several issues that the authors might consider:

  1. Very strange having the Methods section after the Discussion – this should be moved.

Response: The order of sections has been guided by the template provided by the journal. We are obliged to follow this formatting requirement, so we have not made this change.

  1. In the abstract, make clear the %ages are lifetime risk.

Response: Thank you for pointing out this omission – we have added ‘lifetime’ to the percentages in the Abstract (lines 34 and 35).

  1. In the background, it would be helpful to know if 5% to 10% were realistic clinical scenarios – these seem a bit low for e.g. BRCA2 mutations for a 45 year old woman.

Response:  This is now a realistic clinical scenario. Undertaking surgery on the basis of family history alone in the absence of a known mutation (at lower than BRCA levels of risk) has been clinical practice in the UK and other countries for many years [1,2]. Risk reducing salpingo-oophorectomy (RRSO) is now recommended and being undertaken clinically at a greater than 5% threshold of lifetime ovarian cancer (OC) risk. We have shown that RRSO is cost-effective for above these levels of OC risk [3-6]. RRSO is being undertaken for a number of genes other than BRCA1 and BRCA2. These include, RAD51C, RAD51D, BRIP1, PALB2, MLH1, MSH2, MSH6. BRIP1 (5.8% life time risk) and PALb2 (5% lifetime risk) are intermediate risk genes. As are MMR genes (life time risk ~10%), RAD51C (11% lifetime risk) and RAD51D (13% life time risk) [7-9]. RRSO for these genes is part of routine clinical protocols in our and other centres. Additionally more complex models using SNP profiles in combination with other epidemiological and genetic risk factors are being validated which will provide absolute life time risk estimates in these ranges in the not too distant future [10].

We have further edited the introduction to reflect these issues (page 2-3, lines 84-85, 87-98):

Traditionally the most common group of women undergoing surgical prevention have been BRCA carriers, who have a 17-44% lifetime risk of ovarian cancer [26, 27]. [26, 27]. In the UK, women with an estimated lifetime ovarian cancer risk of greater than 10%, who have completed their families, have traditionally been offered risk-reducing surgery [28]. Undertaking surgery on the basis of family history alone in the absence of a known mutation (at lower than BRCA levels of risk) has thus been clinical practice in the UK and other countries for many years [27, 29].  Recently, the 10% threshold was relaxed to 4-5% [14, 15]. A number of new ovarian cancer risk genes have been identified such as RAD51C (life time risk 11%) [30], RAD51D (life time risk 13%), PALB2 (life time risk 5%) [31] and BRIP1 (life time risk 5.8%) [32] testing for which is part of routine clinical practice. RRSO is now offered and being undertaken for these CSGs too. Thus a number of clinical teams now offer RRSO to women in the ‘intermediate’ risk category (5-10%) as well as those in the ‘high’ risk category (over 10%) [28]. Additionally more complex models using SNP profiles in combination with other epidemiological and genetic risk factors are being validated which will provide absolute lifetime risk estimates in these ranges in the not too distant future [12].

  1. The sample is likely to be highly selective in terms of interest of this topic, so this caveat should be highlighted more in the discussion. More information on how the women were approached (e.g. where they on a bank of potential responders from the survey organisation, or where they recruited specifically for this study?  How much were they paid?)  On the positive side, there was a decent spread of educational achievement, which is often a weakness of studies using this method.

Response: We have added the self-selected nature of the sample as a limitation to the discussion (line 300):

The sample was self-selected and may have had greater interest in the topic than the wider general population.

In relation to the query about how women were approached, we have added this information to the Overview section (lines 349-354):

An email containing a web link was sent to SSI panellists who fit the study criteria with respect to gender and age, inviting them to take part. The email did not contain information about the topic of the study. Those responding were directed to a short screening questionnaire. Eligible participants were then presented with a consent form. Incentive points which can be exchanged for shopping vouchers are awarded to SSI panellists for their time (equivalent to ~£0.50 for this 10 minute study).

  1. There is also a related issue of demand characteristics of the study, which could be taken as implying that a response was appropriate.

Response: we acknowledge that any experimental study runs the risk of demand characteristics. We have added this line to the limitations (line 302-304):

Finally, as with any experimental study, we are unable to rule out the possibility of demand characteristics i.e. participants responding in a way they think is expected according to their perceptions of the aim of the study. 

  1. How strong was evidence that “SNPs” was understood?

Response: We did not measure knowledge so we cannot be sure how much of the information women understood. We have added this to the limitations section (lines 291-292):

In addition, we did not assess understanding of the information provided and it is possible that some concepts (e.g. SNPs) may not have been well-understood.

  1. I would like to see the means for EPPM measures reported for each condition, and compared, e.g. was there a greater increase in response efficacy for the SNPs + lifestyle v rare mutation groups? That is, was attributing different cause related to how successful different treatment options were viewed?

Response: Thank you for this suggestion. We have added a table with EPPM means for each experimental group to the Supplementary material and referred to it in line 204.

  1. Were EPPM measures collected only for one group, i.e. are similar measures collected from other groups? And were these measures collected for surgery as an outcome only?  If data collected, it would be useful to see it presented.

Response: EPPM measures were collected from all three experimental groups, with response efficacy questions related to risk reducing surgery and regular TVU screening. Self-efficacy was assessed for surgery only. We hope the addition of the table of means (see previous comment) will make this clearer.

  1. There appeared to be a trend of increasing age being less associated with surgery - this was apparently under-powered due to low numbers of older women. This could be further investigated or examined further.  Would 75 year old women really be considering surgery (or having it recommended?)

Response: Thank you for this observation.  We have now added the following text in the Discussion (line 323-326):

In addition, there were relatively few women in the oldest age group (71-75 years) in this study, and so it is possible the apparent trend of increasing age being less associated with interest in surgery was due to the study being under-powered. This could also be a topic of investigation in future research.

  • Two implications that could be flagged up more. One, that %age risk was not important for surgery decisions, whereas EPPM variables were – this flags up the key importance of consideration of these psychological factors in any future rollout – as well as shows yet again that people struggle to make sense of absolute risks (is 10% good or bad?)

Response: We agree this is interesting and had mentioned this in the Discussion but have now added to this to emphasise that this has implications for possible future rollouts (lines 263-5)). 

The observation that psychological variables had a greater impact on intentions than the absolute risk numbers suggest that this might be important to consider in any potential future national rollouts.

  • Two, that the presentation of material as due to genetic factors v lifestyle appears important in affecting treatment decisions – which relates to the older Marteau vignette studies on this topic. It might be useful to relate the findings to this previous literature, which investigated attributions of cause and genetic determinism over many studies.

Response: We agree the older Marteau vignette studies are interesting, but our study wasn’t set up to be able to address the differences between genetic and non-genetic information in terms of deterministic or fatalistic responses because we did not include a non-genetic condition in our study design. We have therefore added the following information in the Discussion (page 9 lines 269-272):

In previous research using hypothetical scenarios, there has been some evidence to suggest that genetic information leads to more deterministic responses than non-genetic information [37, 42, 52-54]. Our study did not include a non-genetic condition and so does not speak to this aspect of how people respond to personal genetic versus non-genetic information. 

Reviewer 2 Report

First, I apologize for my English. These comments come from a non English speaker.

- Overall I find this study quite interesting and topical. We can see that there would be a rather good adherence to the RRSO if women in the general population were informed that they had a risk of ovarian cancer of 5% or more (whatever the reason). But for me, the idea of "imagining that you are at risk" or "actually being at risk" is conceptually not the same thing. By really being concerned about a risk, women could really respond in a different way. Nevertheless, this study has the merit of having been carried out and still gives trends.

- I wonder about the recruitment of the 1,000 women who completed this questionnaire online. Was the sample representative of women in the general population? Who are these women who had access to this questionnaire (line 297)? We would like to know. Could there be a selection bias? Truly random sampling?

- I find that the information given to the women in the questionnaire is unfortunately too succinct, both in terms of the consequences of RRSO (lines 67-69 of the supplementary materials) and the severity of the disease (lines 17-18 of the supplementary materials). Also, the manuscript and the women's questionnaire do not mention the risk of surgery or anaesthesia at any point in time. It could have been mentioned.

- They do raise the question of the presence of a family history. But the information concerning this family history is missing and it seems to me that we need to know how many women have reported the presence of cancer in the family, or at least the presence of a history of ovarian cancer. These women should not have been excluded from the study if they represent a large number (which is not the case: line 311, Inclusion Criteria)? Because it seems to me that once you know what this disease represents, the perception of risk can be very different. In general, in the appendix put the proportion of women with familial history of cancer and which cancers could have been useful information.

- We do not see in the manuscript the raw results of Q4 of the questionnaire. It would have been interesting to have them: for how many women is the 5% risk of ovarian cancer very different from 2%? (As with other questions related to this part of the questionnaire by the way: Q7 to Q12; that would be a lot of data, but they could have been put in the appendix).

- In the introduction, lines 83 and 84, reference could be made to the lack of effectiveness of screening (pelvic ultrasound and CA-125, which are then taken up in the next paragraph). When the word efficacy is mentioned, we are talking about increased survival. Anual transvaginal ultrasound and CA-125 dosage have not shown any efficacy on survival.

- Also, concerning lines 261 and 262, I am not sure that it is useful to mention the problem of women who would be affected by an RRSO recommendation before the age of 35. Objectively, this concerns almost nobody. In the same way, concerning the point raised on lines 278 to 281: for me it would only concern women with pathogenic variant in BRCA1 or BRCA2 for example, and it seems to me that this research exists. In the general population, there will be no recommendation for RRSO for women younger than the sample analysed (45-75 years old).

- Finally, line 167 (figure 3), the green legend is missing and in table 2, a "Ref" is missing in "Experimental condition".

- One last thing, but this would have been important when designing the questionnaire: would the women need psychological support as part of this RRSO discussion.

Author Response

Reviewer 2

First, I apologize for my English. These comments come from a non English speaker.

- Overall I find this study quite interesting and topical. We can see that there would be a rather good adherence to the RRSO if women in the general population were informed that they had a risk of ovarian cancer of 5% or more (whatever the reason). But for me, the idea of "imagining that you are at risk" or "actually being at risk" is conceptually not the same thing. By really being concerned about a risk, women could really respond in a different way. Nevertheless, this study has the merit of having been carried out and still gives trends.

- I wonder about the recruitment of the 1,000 women who completed this questionnaire online. Was the sample representative of women in the general population? Who are these women who had access to this questionnaire (line 297)? We would like to know. Could there be a selection bias? Truly random sampling?

Response: We thank the reviewer for these thoughtful comments. We acknowledge the possibility of intention behaviour gap which is well described in the literature. We have referred to this in the discussion (line 282-3).

The presence of a potential intention behavior gap is well established for other clinical interventions, and cannot be excluded here. 

We have highlighted the issue of selection bias and limitations to generalisability (as reflected in responses to reviewer 1) (line 300-302).

The sample was self-selected and may have had greater interest in the topic than the wider general population; the generalizability of the findings is therefore uncertain.

- I find that the information given to the women in the questionnaire is unfortunately too succinct, both in terms of the consequences of RRSO (lines 67-69 of the supplementary materials) and the severity of the disease (lines 17-18 of the supplementary materials). Also, the manuscript and the women's questionnaire do not mention the risk of surgery or anaesthesia at any point in time. It could have been mentioned.

Response: We acknowledge this issue – in trying to avoid over-burdening participants with too much reading, we may not have included enough information on some aspects, as mentioned. We have now mentioned this as a limitation. We previously did not find any difference in behaviour outcomes between use of gist and extended versions of decision aids in relation to ovarian cancer risk management [16]. (Lines 308-314).

The information provided before being exposed to the hypothetical risk scenario on ovarian cancer, risk factors and the efficacy of risk management was necessarily basic and brief which is a limitation of this research. However previous research did not find any difference in behaviour outcomes between use of gist and extended versions of decision aids in relation to ovarian cancer risk management [56]. Future research might usefully provide more detailed information containing details about the efficacy of a particular risk management behaviour to encourage ‘danger control’ cognitive processing.

- They do raise the question of the presence of a family history. But the information concerning this family history is missing and it seems to me that we need to know how many women have reported the presence of cancer in the family, or at least the presence of a history of ovarian cancer. These women should not have been excluded from the study if they represent a large number (which is not the case: line 311, Inclusion Criteria)? Because it seems to me that once you know what this disease represents, the perception of risk can be very different. In general, in the appendix put the proportion of women with familial history of cancer and which cancers could have been useful information.

Response: Thank you for raising this important point. We mention in the inclusion criteria (line 368) that eligible women had ‘no previous history of breast or ovarian cancer diagnosis’. We did not collect information about family history of these specific cancers. Overall 621 participants (60.19%) reported having a family history of cancer. We have added this information to Table 1.

- We do not see in the manuscript the raw results of Q4 of the questionnaire. It would have been interesting to have them: for how many women is the 5% risk of ovarian cancer very different from 2%? (As with other questions related to this part of the questionnaire by the way: Q7 to Q12; that would be a lot of data, but they could have been put in the appendix).

Response: We have added responses to these questions by exposure group to the supplementary material (please see our response to Review 1, point 7).

- In the introduction, lines 83 and 84, reference could be made to the lack of effectiveness of screening (pelvic ultrasound and CA-125, which are then taken up in the next paragraph). When the word efficacy is mentioned, we are talking about increased survival. Annual transvaginal ultrasound and CA-125 dosage have not shown any efficacy on survival.

Response:

We agree that annual screening in high risk women is not effective. We have previously shown a benefit of more frequent (4 monthly) biomarker based screening in high risk women using the Ca125 based ROCA algorithm [17]. This leads to a significant stage shift which can be a surrogate for improved survival. However, studies in high risk women are non-randomised and not designed or powered to establish a survival impact. A conclusive mortality impact of screening for ovarian cancer in a low risk population has not yet been established [18]. Evaluation of long term follow up of the UKCTOCS cohort for a delayed mortality impact is ongoing to further assess this. We have further edited the manuscript to better reflect these issues. We have edited the introduction to include these points (lines 105-7):

This 4 monthly longitudinal CA125 biomarker driven surveillance strategy using the risk of ovarian cancer (ROCA) algorithm, may be beneficial in women at high risk of ovarian cancer [34]. We have shown that this is associated with a significant stage shift, which can be a surrogate for improved survival [35].

- Also, concerning lines 261 and 262, I am not sure that it is useful to mention the problem of women who would be affected by an RRSO recommendation before the age of 35. Objectively, this concerns almost nobody. In the same way, concerning the point raised on lines 278 to 281: for me it would only concern women with pathogenic variant in BRCA1 or BRCA2 for example, and it seems to me that this research exists. In the general population, there will be no recommendation for RRSO for women younger than the sample analysed (45-75 years old).

Response: The reviewer highlights an important issue. Most women from the general population who are at increased risk will fall in the intermediate risk category [19]. RRSO at intermediate OC risk levels is recommended to be undertaken over the age of 50, particularly where this is not associated with a high risk gene mutation [6]. Even for intermediate risk genes like BRIP1 it is recommended at older ages over 45, (not younger ages as for BRCA genes) [4,6]. This study predominantly explores intentions at risk levels below the traditional high penetrance genes. We have made some edits to the discussion section (lines 297-300):

However, most women from the general population who are at increased risk of ovarian cancer will fall in the intermediate risk (5-10% lifetime risk) category [9].  RRSO at intermediate OC risk levels (including for moderate penetrance CSGs) is recommended to be undertaken over the age of 45-50 years [15]. 

- Finally, line 167 (figure 3), the green legend is missing and in table 2, a "Ref" is missing in "Experimental condition".

Response: Thank you for spotting this – we have now added the green legend and the extra ‘Ref’.

- One last thing, but this would have been important when designing the questionnaire: would the women need psychological support as part of this RRSO discussion.

Response: Thank you for highlighting this important issue. Offering psychological support for those who need it as part of the RRSO discussion is already part of clinical practice. In our practice we routinely offer this to women who may need it. We have a psychologist embedded in our multidisciplinary clinical team managing high risk women. This is part of standard protocols for risk reducing mastectomy too. In our experience a small but not insignificant proportion of women need psychological support around RRSO. A recent paper from our group highlights women’s desires and needs for psychological support around RRSO decision making (paper in press). We have made edits to

  Offering psychological support for those who need it as part of the RRSO discussion and decision-making process is part of routine clinical practice in many centres today. Our study highlights the importance of incorporating this into future national guidelines. 

Reviewer 3 Report

Manuscript cancers-973199 entitled, “Women’s intentions to engage in risk-reducing behaviours after receiving personal ovarian cancer risk information: an experimental survey study” was reviewed.

The authors conducted a cross-section study using questionnaire survey including six items to evaluate the intent to have risk-reducing surgery or other risk-management option on the receipt of the result of ovary cancer risk stratification (SNP, BRCA, or lifestyle factors). They correlated the intent to treatment with demographics, perceived risk and severity of the disease, self-efficacy and perceived response efficacy. Totally, 1,017 women of a general population were recruited.

The following comments were provided. 

  1. The authors should clearly describe the purpose and hypothesis of the study.
  2. In the conclusion, how do the results translate into action in plan of a population-wide screening of ovary cancer?
  3. A retrospective case control study or a prospective cohort study to compare the cost-effectiveness and outcome between screening in a high risk group and in a general population may provide more evidences to support a population-wide screening of ovary cancer.
  4. The section of Introduction is redundant.
  5. Several areas with unclear meaning need be rewritten, such as line 81-85, line 107-111, line 118-120, line 147-149, line 213-219.
  6. A comma is needed between “chage” and “data” in line 69, “cancer” and “less” in line 109.

Author Response

Reviewer 3

Manuscript cancers-973199 entitled, “Women’s intentions to engage in risk-reducing behaviours after receiving personal ovarian cancer risk information: an experimental survey study” was reviewed.

The authors conducted a cross-section study using questionnaire survey including six items to evaluate the intent to have risk-reducing surgery or other risk-management option on the receipt of the result of ovary cancer risk stratification (SNP, BRCA, or lifestyle factors). They correlated the intent to treatment with demographics, perceived risk and severity of the disease, self-efficacy and perceived response efficacy. Totally, 1,017 women of a general population were recruited.

The following comments were provided. 

  1. The authors should clearly describe the purpose and hypothesis of the study

Response: At the end of the Introduction we stated the purpose of the study as follows:

Our specific aims were to: (1) explore whether women in the general population are willing to undergo ovarian cancer risk assessment which includes genetic testing; (2) examine whether women’s potential acceptance of risk-reducing surgery differs depending on whether their estimated risk is 5% or 10%; (3) examine whether women’s potential acceptance of risk-reducing surgery differs depending on whether their estimated risk is based on a single rare genetic variant of high penetrance or a more complex combination of genetic and non-genetic factors. We also explored whether threat and efficacy cognitions mediated any observed between-group differences, and examined the associations between these cognitions (threat, efficacy) and acceptance of risk-reducing surgery in the sample overall.

  1. In the conclusion, how do the results translate into action in plan of a population-wide screening of ovary cancer?

Response: We thank the reviewer for raising this important point. We have undertaken significant research in the area of population screening for cancer genes [15,20-22] and also specifically for ovarian cancer risk. Our pilot study shows that population testing for life time risk of ovarian cancer is feasible, acceptable, has high satisfaction does negatively affect quality of life/psychological well-being in general population women [10]. It provides a new paradigm for ovarian cancer prevention and can prevent thousands more cancers than the current clinical approach.[23] There is now need for large implementation studies with long term outcomes to provide real world evidence and develop models for implementing this approach in general population women. We are in the process of developing such studies. We have made edits to the discussion (lines 340-344):

Population testing provides a new paradigm for ovarian cancer prevention and can prevent thousands more cancers than the current clinical approach [57]. Jewish population studies support population testing for CSGs [58]. Our pilot study shows that population testing for life time risk of ovarian cancer is feasible, acceptable and has high satisfaction in general population women [12].

  1. A retrospective case control study or a prospective cohort study to compare the cost-effectiveness and outcome between screening in a high risk group and in a general population may provide more evidences to support a population-wide screening of ovary cancer.

Response:  We have shown that population testing for breast and ovarian cancer gene mutations is extremely cost-effective compared to the current clinical approach [24-26]. We recently evaluated the potential for population testing for BRCA genes and showed it was cost-effective across multiple countries and health systems [27]. We agree that what is needed is prospective implementation real world studies evaluating long term impact and a range of outcomes for population risk stratification in general population women.[23,28,29] Additional cost-utility analysis will be an outcome of the implementation studies. We have added the following text to the Discussion (lines 344-48):

However, there is now need for large implementation studies with long term outcomes to provide real world evidence and develop context specific models for implementing this approach in general population women. This will valuably inform future policy decisions regarding population-wide risk stratified approaches for risk adapted ovarian cancer screening and prevention.

  1. The section of Introduction is redundant.

Response: We are not clear what section this refers to and so have not made any edits in response to this comment at this time. 

  1. Several areas with unclear meaning need be rewritten, such as line 81-85, line 107-111, line 118-120, line 147-149, line 213-219.

Response: We have now edited these sentences to improve their clarity.

  1. A comma is needed between “change” and “data” in line 69, “cancer” and “less” in line 109.

Response: Thank you – these have been added.

Round 2

Reviewer 3 Report

Revised manuscript cancers-973199 entitled, “Women’s intentions to engage in risk-reducing behaviours after receiving personal ovarian cancer risk information: an experimental survey study” was reviewed.

The authors conducted a cross-section study of 1,017 women using questionnaire survey to evaluate the intent to have risk-reducing surgery or other risk-management option on the receipt of one of three positive result of ovary cancer risk stratification (SNP, BRCA, or lifestyle factors). They correlated the intent to treatment with demographics, perceived risk and severity of the disease, self-efficacy and perceived response efficacy. They aim to evaluate the public acceptability of such approaches.

The following comments on the revised version were provided.

  1. For the purpose (1) line 155-156: Whether the study group represents the general population remains to be determined. The high percentage of family cancer history and experience of cervical and breast cancer screening suggest the selective bias, i.e. they more intend to have a screening test.
  2. For the purpose (2) & (3) line 156-160: The comparisons need be made among three groups in the Table 1 for the demography, in the Figure 3 for the management chosen. Besides, there was only one column in the Figure 2.
  3. Table 2 lack column for other behavioral response such as to have surveillance. The data of extended parallel processing model variable is blank The number and percentage of experimental condition is incorrect, when refer to the Figure 3 and the text.
  4. In a study, there must be a hypothesis to be tested and concluded with the results, not simply described the results. The authors did not discuss how to translate the more self-efficacy and perceived response-efficacy affect the decision to undergo risk-reducing surgery into the screening planning and clinical practice.
  5. The Introduction section becomes even longer, which makes the article not readable. Some grammar problem exists, such as the sixth paragraph. The authors should try to shorten it. Some could be moved to the Discussion.
  6. Whether the hypothetical scenario of virtual subjects represent the true world is uncertain. It may be a study design flaw.
  7. Line 107: Early cancer detection via screening resulted in stage shift and improved survival, but no imply reduced mortality.
  8. The “5% lifetime risk due to SNPs and lifestyle factors” and the “10% lifetime risk due to SNPs and lifestyle factors” should be clearly defined. The understanding of the meanings by volunteers affects the results.
  9. The results maybe different between childbearing age and peri- or post-menopausal age.